# Stimuli-Responsive Drug Delivery Systems for the Diagnosis and Therapy of Lung Cancer

**DOI:** 10.3390/molecules27030948

**Published:** 2022-01-30

**Authors:** Xu Lin, Jiahe Wu, Yupeng Liu, Nengming Lin, Jian Hu, Bo Zhang

**Affiliations:** 1Department of Thoracic Surgery, The First Affiliated Hospital, Zhejiang University School of Medicine, Hangzhou 310003, China; linxu_001@zju.edu.cn; 2Key Laboratory of Clinical Cancer Pharmacology and Toxicology Research of Zhejiang Province, Affiliated Hangzhou First People’s Hospital, Zhejiang University School of Medicine, Hangzhou 310003, China; jill_jh@zju.edu.cn (J.W.); liuyp0503@zju.edu.cn (Y.L.); lnm1013@zju.edu.cn (N.L.); 3Cancer Center, Zhejiang University, Hangzhou 310003, China

**Keywords:** stimuli-responsive, drug delivery, lung cancer

## Abstract

Lung cancer is the most commonly diagnosed cancer and the leading cause of cancer death worldwide. Numerous drugs have been developed to treat lung cancer patients in recent years, whereas most of these drugs have undesirable adverse effects due to nonspecific distribution in the body. To address this problem, stimuli-responsive drug delivery systems are imparted with unique characteristics and specifically deliver loaded drugs at lung cancer tissues on the basis of internal tumor microenvironment or external stimuli. This review summarized recent studies focusing on the smart carriers that could respond to light, ultrasound, pH, or enzyme, and provided a promising strategy for lung cancer therapy.

## 1. Introduction

Lung cancer is the most diagnosed cancer in the world, with over two million new cases in 2020 (Figure 1) [1]. The current therapeutics for lung cancer mainly include surgery, drug therapy, and radiotherapy [2]. Indeed, the development of either small-molecule anticancer drugs or biologics, such as monoclonal antibodies for immunological targets has dramatically increased the clinical efficacy for the treatment of lung cancer in recent years [3]. However, problems remain unsolved, such as relatively low clinical response rates of monoclonal antibodies, unwanted adverse effects of targeting drugs, and drug resistance after a duration of exposure to certain agents [4,5]. To further enhance the drug effectiveness, one important strategy is to deliver drug cargoes specifically into lung cancer tissue using tailored carrier systems, which would confer ‘desirable’ properties to therapeutic agents and meanwhile compensate significant drawbacks in clinical applications [6,7].

Drug delivery systems are generally composed of carriers and therapeutic agents, and the conjugation of these two components would improve drug stability, desirable drug accumulation, and drug release, thus providing a promising strategy for lung cancer therapy [8]. Recently, intelligent drug carrier systems could precisely deliver drugs on the basis of unique tumoral microenvironments or external stimulus, such as pH, enzymes, reactive oxygen species (ROS), photodynamic, and so on [9,10,11,12,13]. Those stimuli-responsive drug carriers such as micelles [14], liposomes [15], hydrogels [16], and nanoparticles [17] have been fabricated and undergo cleavage of chemical bonds or conformational change to release drugs at a specific pattern.

With the deep understanding of lung cancer studies, researchers have realized the unique tumor microenvironment that occurs during the progression of lung cancer, such as acidic tumoral environment [18], elevated intracellular glutathione (GSH) conditions [19], and high levels of ROS [20], thereby provides an opportunity for the application of drug delivery systems for lung cancer therapy. In addition to stimuli-based nanocarriers that are sensitive to internal (temperature, pH, redox, enzymes reactions) environment, external (ultrasound, electric field, light, magnetic field) conditions could also be utilized to trigger fast drug release at a designated localization [21,22,23,24]. Additionally, nanocarriers can be organic, inorganic, or hybrid and follow either an active mechanism or a passive mechanism for tumor targeting.

Herein, we reviewed the recent studies on stimuli-responsive drug delivery systems for the diagnosis and therapy of lung cancer. These drug delivery systems could respond to pH, enzyme, ROS, magnetic field, photodynamic, or ultrasound, aiming to enhance drug efficacy and meanwhile minimize side effects (Table 1).

## 2. Light-Responsive Nanocarriers

Because of its relative safety and noninvasive character, light has been widely applied for remotely controlled drug delivery [25]. Short-wavelength light, including ultraviolet (UV) and visible light, can be utilized to destruct photolabile groups directly for on-demand drug release. However, their limited penetration ability hinders their biomedical application. In contrast to short-wavelength light, near-infrared (NIR) light (780–2500 nm) is able to penetrate deeper through the tissues, and this feature of NIR is preferable for remote control of the desired drug release. To take advantage of both short-wavelength and NIR light, up-conversion nanoparticles (UCNPs), which are capable of transferring NIR light to short-wavelength light, are utilized to fulfill both deep penetration and short-wavelength light-responsive drug release. For example, Ming-Fong Tsai et al. used a UV-responsive o-nitrobenzyl ester (ONB) containing amphiphilic block copolymer to construct a polymersome [26]. Then core-shell UCNPs and doxorubicin (DOX) were co-encapsulated to the polymersome, enabling NIR light-inducing photolysis and on-demand drug release for enhanced chemotherapy of lung cancer.

Apart from the immediate breakdown of photolabile groups and structural transformation of the therapeutic system triggered by external light sources, some nanoparticles can be activated by light to generate reactive oxygen species (ROS) or increase local temperatures, which can be applied in killing tumor cells [27,28]. These therapeutic strategies are known as photodynamic therapy (PDT) and photothermal therapy (PTT), respectively. For instance, the core-shell gold nanocage@manganese dioxide (AuNC@MnO_2_) nanoparticles were developed by Ruijing Liang et al. to simultaneously ablate primary triple-negative breast cancer and prevent lung metastases via oxygen-boosted immunogenic PDT [29]. Gold nanomaterials can be utilized not only as PDT agents but also PTT agents. Jianli Chen et al. fabricated titania-coated gold nanobipyramids and loaded anticancer drug combretastatin A-4 phosphate to induce synergistic chemotherapy and PTT to lung cancer with 1064 nm laser irradiation [30].

Accumulating evidence indicates that the combining of various therapeutic modalities (e.g., PTT, PDT, immunotherapy, chemotherapy) is a promising strategy for enhanced anti-tumor efficiency with minimized side effects. Additionally, the ROS or thermal effects generated during PDT or PTT can also trigger the drug release in the meantime. For instance, afatinib was loaded into the poly(l-lysine)-conjugated chlorin e6 (Ce6) derivative nanoparticle and covered by hyaluronic acid (HA) shells by Man Zhang et al. [31]. Upon NIR irradiation, the photosensitizer Ce6 generated ROS which induced the oxidation of the thioether linker and subsequently triggered the release of afatinib for improved therapeutic efficacy in non-small-cell lung cancer (NSCLC) treatment. Similarly, Chongchong Wang et al. fabricated a palladium nanosheet (PdNS) to carry carbon monoxide via reaction with transition metals, which could be destroyed by the heat generated from the PdNS upon 808 nm laser irradiation [32].

Among multiple therapeutic modalities, immunotherapy is now regarded as the first-line therapy for many cancer indications and revolutionized the field of oncology during the past decade [33]. Specifically, it has been demonstrated that phototherapy can trigger immunogenic cell death (ICD) of tumor cells [34]. However, with the immunosuppressive tumor microenvironment, as well as multiple mechanisms involved, adaptive immune resistance may restrain the anti-tumor activity of the ICD cascade. Thus, immunotherapeutic molecules are sometimes introduced into the delivery system for enhanced immunotherapy-assisted synergistic treatment. Additionally, light-controlled release can further raise the specificity [35,36]. For instance, Jingchao Li et al. proposed a second near-infrared (NIR-II) photothermal immunotherapy using a semiconducting polymer nanoadjuvant (SPNIIR), which was composed of a semiconducting polymer nanoparticle core, a toll-like receptor agonist R848, and a thermally responsive lipid shell (DPPC) [37]. Under the irradiation of a NIR-II laser, the thermal effect of the semiconducting polymer nanoparticle core caused the removal of the DPPC shell and induced the on-demand release of R848. Consequently, the synergistic photothermal immunotherapy could suppress primary tumors and eliminate lung metastasis in vivo (Figure 2).

Light-triggered drug release can also be combined with other stimuli, including internal stimuli (e.g., pH [38,39,40], enzyme [41], glutathione [42,43], and other external stimuli (e.g., radio frequency [44]), for enhanced targeted therapy to lung cancer. For example, as the tumor microenvironment is enriched in esterase, introducing the esterase-labile ester bond to the therapeutic system can achieve tumor microenvironment-responsive drug release in situ. Combined with the photoexcited effect, the encapsulated drug is able to be released to its maximum extent [45]. Similarly, introducing a disulfide bond to a photosensitive therapeutic system can fabricate a dual-responsive (GSH/light) therapeutic agent as well [42]. Radio frequency stimulation is also a widely adopted noninvasive therapeutic tool by generating heat like laser irradiation. Based on this concept, Animesh Pan et al. loaded DOX and iron oxide nanoparticles (IONs) coated with a gold nanoshell into the layersome to perform combined hyperthermia and triggered drug release via radio frequency or NIR stimulation. Compared to the single PTT or radio frequency treatment, the DOX and nanoparticles-loaded layersome with dual stimulation displayed a higher therapeutic effect on non-small cell lung cancer A549 cells [44].

Light-responsive therapeutics exhibit multiple advantages as mentioned above and hold great potential for clinical treatment of lung cancer [41]. Furthermore, molecules that absorb light and generate heat or ROS sometimes can also emit fluorescence or transfer the heat to photoacoustic (PA) signal and thus are employed for imaging diagnosis during the therapeutic process. For instance, indocyanine green (ICG) is a typical photothermal molecule and can also be utilized for NIR fluorescence and PA imaging [43]. Other fluorescent molecules, such as IR780 [46,47], Cy7 [48], and NIR770 [41], are also applied as theranostic agents to the treatment of lung cancer. Ziying Li et al. [48] established a chitosan-based nanocomplex CE7Q/CQ/S to deliver molecular-targeted drug erlotinib (Er), Survivin shRNA-expressing plasmid (SV), and Cy7 for simultaneous NIR fluorescence imaging and monitored chemo/gene/photothermal tri-therapies therapy for NSCLC bearing epidermal growth factor receptor (EGFR) mutations. With the guidance of NIR imaging, the therapy was more accurate, and the therapeutic outcome was able to be observed in real-time.

## 3. Ultrasound-Responsive Nanocarriers

Ultrasound (usually defined as > 20 kHz) is widely used for diagnosis and therapy in the clinic [49]. Due to its merits in clinical application, including cost-effectiveness, simplicity, and particularly noninvasiveness, ultrasound has been adopted as an external stimulus for smart therapeutics to trigger amplified therapeutic effects. Micro-/nanobubbles, liposomes, liquid perfluorocarbon droplets, micelles, or mesoporous silica nanoparticles (MSN) have been developed as ultrasound-responsive drug carriers after rational design and synthesis.

The fundamental mechanisms underlying ultrasound-mediated therapy mainly include thermal effect, mechanical effect, and chemical effect [50]. The thermal effects are attributed to acoustic energy produced by propagating ultrasound. Surrounding biological tissues can absorb part of the energy and thus lead to a temperature increase in the respective areas. Relative high temperature is able to kill cancer cells directly, while hyperthermia (above 80 °C) caused by high intensity focused ultrasound (HIFU) may be associated with undesired complications, such as second- and third-degree skin burns [51]. On the other hand, the temperature increase induced by ultrasound could probably trigger the thermal instability of the drug delivery system and enable targeted controlled drug release.

The mechanical effects are mainly generated from ultrasound pressure, acoustic streaming, and ultrasound-induced oscillation or cavitation [49], among which cavitation is often leveraged for drug delivery owing to its specific influence on biological processes. For example, upon exposure to intense ultrasound energy, perfluoropentane containing nanobubbles would go through rapid bubble destruction triggered by rapid contraction and expansion of the bubbles. The bubble shell can be weakened by this mechanical stress until the bubble ruptures, leading to the drug release in situ [52]. Sonoporation is the process of pore formation in a cell membrane upon exposure to ultrasound and belongs to one of the cavitation effects, which could facilitate the intracellular transport of drugs. Moreover, cavitation can also widen the interspace between endothelial cells and thus enhance the penetration into adjacent tissues [50]. However, unwanted cavitation effects may take place in the presence of residual air bubbles, and thus the implementation of ultrasound in the treatment of lung cancer would lead to undesired drug release in the process of drug transportation.

The chemical effects of ultrasound mediated treatment can also be called sonodynamic therapy (SDT). Oxygen, sonosensitizer, and appropriate ultrasound are three necessary components to complete the process of SDT, and the generation of ROS upon focused ultrasound exposure can cause site-specific profound damage to tumor tissues [53]. The combinatorial treatment of SDT with other therapeutic strategies, such as chemotherapy and chemodynamic therapy (CDT), has a synergistic effect in the treatment of lung cancer [54]. For example, Shiyan Fu et al. synthesized PEGylated Co_2_Fe_2_O_4_ nanoflowers (CFP) [55]. This CFP occupying multivalent elements (Co^2+/3+^and Fe^2+/3+^) exhibited strong Fenton-like and catalase-like activity. Moreover, CFP could also be employed for high-performance SDT as a brand-new sonosensitizer attributed to the ultrasound-triggered electron (e^−^)/hole (h^+^) pair separation from the energy band. After efficient accumulation in the tumorous region as revealed by magnetic resonance imaging, CFP could generate ^•^OH for CDT relying on Fenton-like reactions and generate molecular oxygen due to the catalase-like activity which may promote the production of ^1^O_2_ for SDT. Combined SDT/CDT could further efficiently trigger ICD and thus synergistically suppress primary and distant tumors, as well as lung metastasis.

Thermal, mechanical, and chemical effects are not entirely independent and sometimes two or three of them may result in a synergistically therapeutic modality. For instance, Xiaotu Ma et al. fabricated a cerasomal perfluorocarbon nanodroplet (D-vPCs-O_2_) with an atomic layer of polyorganosiloxane and pH-sensitive tumor-targeting peptide [55]. Oxygen and doxorubicin were co-loaded into the nanodroplets. HIFU was utilized to trigger the release of cargoes and simultaneously enhance ultrasound imaging, therefore achieving imaging-guided drug delivery. Mild-temperature HIFU (M-HIFU) could also be applied to slightly elevate tumor temperature and accelerate tumor blood flow. Consequently, ultrasound-triggered oxygen release and temperature elevation jointly relieved tumor hypoxia and alleviated multiple drug resistance, and these two effects jointly enhanced the drug therapeutic efficacy to lung metastasis [56] (Figure 3).

Two ultrasound parameters, acoustic frequency, and intensity, are often manipulated to induce desired biological effects. For example, Yichen Liu et al. constructed a functionalized smart nano sonosensitizer (EXO-DVDMS) by loading sinoporphyrin sodium (DVDMS), which was an excellent porphyrin sensitizer with both therapeutic and diagnostic features, onto homotypic tumor cell-derived exosomes [53]. A guided-ultrasound (US1, 2 W, 3 min) was first introduced to promote the accumulation of EXO-DVDMS in the tumor region, and subsequently, the therapeutic-ultrasound (US2, 3 W, 3 min) was applied for SDT, thus enhancing the targeted delivery of DVDMS to primary as well as metastatic lung tumors. In addition, other external stimuli, such as magnetic fields, can be incorporated with ultrasound-responsive delivery systems and achieve precisely controlled release. Senay Hamarat Sanlier et al. fabricated liposome-based nanobubbles [57]. Pemetrexed and pazopanib were conjugated with peptide and then attached to the surface of magnetic nanoparticles. After the functionalized magnetic nanoparticles were encapsulated into the liposomes, pemetrexed and pazopanib carrying nanobubble systems with magnetic responsiveness and ultrasound sensitivity were constructed for NSCLC targeted delivery [58]. Furthermore, the inclusion of magnetic nanoparticles can not only enable the magnetic field-guided targeted delivery but also considerably improve both the stability and phase conversion efficiency of nanodroplets. 

## 4. PH-Responsive Nanocarriers

The lactic acid and certain end products produced by lung cancer cells, which are related to an abnormally fast metabolism and proliferation, lead to a more acidic environment (pH 5.7–6.9) in tumor tissues than normal physiological pH (pH 7.4) [59,60]. pH-sensitive nanoparticles could maximize drug release in the pulmonary tumor microenvironment and minimize drug release en route to the tumor, enhancing the accumulation of nanosystems in the tumor tissue and meanwhile improving the reliability and safety of targeted therapy for lung cancer.

Lee and coworkers found that the CHEMS-based liposomes could be efficiently triggered by the acidic pH and these drug-loaded carriers exhibited an outstanding anti-tumor effect in NSCLC. Folate receptor beta (FRβ), which was usually overexpressed in M2 tumor-associated macrophages (TAMs) and NSCLC cells, was associated with the poor prognosis of NSCLC patients [61]. Since cholesteryl hemisuccinate(CHEMS) was unstable in acidic conditions, the conjugation of CHEMS onto PEG-Folate was used to construct pH-sensitive liposomes to achieve targeted drug release. As a result, these liposomes showed a faster drug release profile at pH 6.5 than that at neutral pH, and a burst release was observed at pH 4.0. Acylhydrazone bond could also be used as a pH-sensitive linker between polyethylene glycol (PEG) and tumor-targeted hyaluronic acid (HA), resulting in HA-ERL/BEV-LPH nanoparticles to treat NSCLC. In vitro release profiles of HA-ERL/BEV-LPH nanoparticles demonstrated that pH-sensitive adipic acid dihydrazide (ADH) could effectively control drug release in the acidic pH and release drugs faster than that at physiological pH [62]. The poly-γ-benzyl-l-glutamate and an amphiphilic copolymer d-α-tocopherol polyethylene glycol succinate mixed micellar system could control the release of DOX by changing the secondary structures of poly-γ-benzyl-l-glutamate. The DOX-loaded mixed micelles exhibited great anti-tumor efficacy in human lung cancer A549 cells-bearing nude mice [63].

Additionally, the acid pH of the tumor environment also could trigger charge reversal to promote cellular internalization and nuclear entry in the treatment of lung cancer (Figure 4). Nanocarriers with positive surface charge usually bear short blood circulation half-life due to an unspecific adsorption, whereas the addition of TAT peptide would overcome this drawback by improving drug uptake of tumor cells. Anhydride (DA) groups can be utilized to mask the positive charges of TAT. Once the carrier accumulated in the tumor acidic environment, a charge reversal from negative to positive occurred and the targeting ability of TAT was recovered. Zhou et al. generated a DA-TAT carrier for pH-triggered cell uptake and nuclear targeting, which possessed beneficial effects in treating lung metastasis [64]. Similarity, Zhao et al. developed a pH-responsive poly(histidine) (PHis) based polymer consisting of a cationic lipid core and a triblock copolymer methoxy poly(ethylene glycol)-poly(histidine)-poly(sulfadimethoxine) (mPEG-PHis-PSD or PHD). Acidic pH transformed PSD from a negative to neutral charge, which resulted in a fast dissociation from lipid core, thus achieving tumor-selective accumulation, effective internalization, and efficient anti-tumor activity for NSCLC therapy [65].

Prodrug-based nanosystem was a great choice with high drug-loading capacity and could load additional drugs for synergistic treatment [66,67]. Ma and coworkers constructed a pH-sensitive doxorubicin (DOX) prodrug for lung cancer therapy. The hydrophilic segment U11-PEG was introduced to DOX by pH-responsive PHis. Curcumin (CUR) was loaded into DOX-based nanoparticles as a secondary anti-tumor drug. When the prodrug-based codelivery system U11-DOX/CUR nanoparticles were exposed to the tumor acidic microenvironment, DOX and CUR were released simultaneously because of the protonation of pHis. This study suggested that the U11-DOX/CUR nanoparticles are pH-responsive systems and had a potent anti-tumor effect on lung tumor cells [68]. *Cis*-aconitic anhydride-modified doxorubicin (CAD) was designed for pH-sensitive drug release in another study. CAD showed specific distribution in the tumor tissues after 12 h post-injection, exhibiting excellent lung tumor-targeting ability of these nanoparticles. The acid-responsive cis-aconityl linkage between the cis-aconitic anhydride (CA) and antitumor drug DOX could be hydrolyzed, and the release of DOX would accelerate the linkage breakdown once the nanoparticles reached tumor tissues [69]. 

## 5. Enzyme-Responsive Nanocarriers

Enzymes are essential biomolecules that maintain normal functions of living organisms, e.g., growth, development, metabolism, aging, disease, and immunity. Aberrant enzyme expression was commonly observed in multiple disease-associated microenvironments and cells, especially lung cancer [70,71]. The overexpressed enzymes mainly include matrix metalloproteinases (MMPs), hyaluronidase (HAase), esterase, NAD(P)H, and quinone oxidoreductase1 (NQO1). Enzyme-responsive nanoparticles have attracted considerable attention owing to their selectivity, effectiveness, and rapidity of enzymatic reactions in lung cancer treatment [72,73,74]. For example, MMPs are a family of proteolytic zinc-dependent secreted endopeptidases that can specifically degrade a variety of compositions in extracellular matrices (ECMs). Abnormally high expression of MMPs within lung tumor tissues could be utilized for the development of enzyme-responsive drug delivery systems [72,75]. Based on the concept that MMP-9 and MMP-2 can specifically degrade collagen and basement membrane, gelatin or the MMP-responsive peptides are generally conjugated onto the surface of nanoparticles for targeted lung cancer therapy [76]. Guo et al. conjugated MPEG to PCL using an MMP-2 sensitive peptide linker (GPLGIAGQ), resulting in lung targeted Cur-P-NPs, which exhibited a superior drug release profile than that of the non-responsive control [77]. Likewise, an engineered shell composed of phosphorylcholine (PC) and enzyme-responsive peptides was constructed by Kang and coworkers. The functional vehicle facilitated the precise delivery of loaded protein drugs once reaching the lung tumor sites; then the therapeutic agents would bind to pulmonary tumor cell surface receptors to suppress cell growth [78]. Gianneschi designed an MMP-9-sensitive nanocarrier to deliver the immunotherapeutic small molecule (1V209) for selective immune activation. The aforementioned platform enhanced drug efflux and inhibited in vivo lung tumor metastasis [78]. In addition, gelatin can also be used as a substrate for MMPs, and be incorporated as the carrier skeleton. Verma designed a smart inhalable nanocarrier by complexation of gelatin with cisplatin for lung tumor therapy. After the degradation of gelatin, cisplatin was exposed to physiological salts and exchanged with chloride ions, leading to the fast release of cisplatin at the tumor site [79].

Apart from MMPs, HAase is usually combined with other sensitive patterns to generate dual- or multi-responsive nanoparticles to improve lung cancer therapeutic effectiveness. Hyaluronic acid (HA) was used to construct the hydrophilic shell, while hydrophobic compounds could be introduced into the HA backbone by environmentally responsive bonds. For example, Tang’s group utilized pH-sensitive hydrazone bonds to construct enzyme and pH dual-responsive hyaluronic acid nanoparticles [80]. In a similar study, He and coworkers designed lung cancer cells’ active-targeting, enzyme, and ROS-sensitive nanoparticles named HPGBCA to deliver afatinib for NSCLC therapy. Poly(glycidylbutylamine) (PGBA) was a cationic amphiphilic compound with a ROS-sensitive thioether linker. The anionic HA shell could actively target CD44 receptor-overexpressed tumor cells and mask the positive charge of PGBA for long circulation in the bloodstream. When HPGBCA reached HAase-enriched lung tumor sites, the HA shell was degraded to expose positively charged cores and accelerate the lysosomal escape. Subsequently, the Ce6 of HPGBCA could produce ROS under NIR irradiation to trigger the oxidation of ROS-sensitive linkers for drug release [31].

The esterase-sensitive nanocarrier is also a great choice in enzyme-responsive drug delivery systems for the lung. A nanoparticle named HAPBA, which was designed by Cho and coworkers for lung cancer therapy, would release drugs at esterase-enriched tumor tissue environment. Ester bonds were used to join 4-Phenylbutyric acid (PBA) and HA backbone for the quick release of curcumin and PBA. PBA was not only the hydrophobic segment in the structure of these nanoparticles but also an efficient inhibitor of histone deacetylase (HDAC). The cleavage of ester bonds realized the rapid release of curcumin and PBA, exhibiting efficient tumor growth suppression in lung adenocarcinoma [81]. Similarly, Ren et al. designed a gold nanorod–curcumin conjugate held together by an esterase-labile ester bond. This conjugate showed a rapid and sustained release of curcumin. In the absence of esterase, encapsulated drugs were completely restricted inside the nanoparticles. When the concentration of esterase increased, an abrupt curcumin release was observed, suggesting that ester hydrolysis was an essential trigger of drug release. As a result, the introduction of the ester bond enhanced the inhibitory effects of the nanorod–curcumin conjugate on human lung cancer A549 cells [45].

NQO1 enzyme is a cytosolic reductase that is abnormally overexpressed in multiple cancers, including lung cancer [82,83]. Trimethyl-locked quinone propionic acid (QPA) reacts with NQO1 to form a lactone-based group via intramolecular cyclization. To take advantage of the fundamental features of NQO1, an NQO1-responsive nanoparticle termed QPA-P was designed by Kim et al. for lung cancer therapy. Poly(ethylene glycol) (PEG) was used as the hydrophilic segment and QPA-locked polycaprolactone (PCL), which was conditionally triggered by NQO1, was the hydrophobic tail that imparted amphiphilic property to QPA-P. After a cascade two-step cyclization process with the NQO1 enzyme, the particle size of QPA-P increased and loaded DOX rapidly released into surrounding medium and tumor cells, indicating that NQO1-sensitive micelles were promising for drug delivery in lung cancer [84].

There are still some issues that should be considered. The drug release efficiency totally depends on the specific enzyme concentration of lung tumor cells. Hence, the type and the stage of the tumor must be taken into full consideration. Furthermore, the stability of enzyme-responsive nanoparticles in the physiological environment is also important.

## 6. Discussion and Perspective

In the last ten years, lung cancer therapy has attracted intense attention not only because of the fact that lung cancer is still the most common type of cancer in the world but also considering the fast development of targeted therapy drugs for lung cancer. However, those medicines that have already benefited millions of patients are faced with undesirable adverse effects and thus, cause discontinuation of the treatment, which would substantially undermine the effectiveness of therapeutic drugs. Stimuli-responsive carriers provide a promising solution to overcome the drawbacks of traditional medicine, mainly through enhancing the accumulation of drugs and precisely releasing drugs at lung tumor sites. However, major challenges still remain and hinder the wide application of these drug carriers. Most of these carriers are complicated in terms of structure and formulation, and it is difficult to control the stability, integrity, and physiochemical property in the scale-up production. Additionally, the tumor microenvironment is one of the critical obstacles that devastate the transport of drug carriers and also greatly affect the drug release under certain circumstances, whereas few drug carriers have fully considered the complexity of the lung cancer microenvironment at the time of original design and development. In spite of these current hurdles, scientists are making tremendous efforts to fabricate drug carriers and optimize their features.

It is worth mentioning that several stimuli-responsive systems have already been approved by the FDA and are currently under clinical trials. For example, the first heat-activated nanocarrier (lyso-thermosensitive liposomal doxorubicin, LTLD, ThermoDox^®^) that was utilized in clinical trials has shown a promising advantage in improving the overall survival of primary liver cancer patients [85]. Some other systems, such as ferumoxytol and oral iron, which could be used for the treatment of iron deficiency anemia in the setting of chronic kidney disease and enhanced MRI, have shown their clinical applications in therapeutic functions and cancer diagnosis [86]. An intelligent, target-controlled liposome formulation (LiPlaCis^®^) has been evaluated for its therapeutic effects in prostate cancer treatment [87]. These inspiring results elicited that the exploration of unique targeted biomarkers was necessary to utilize interior stimulus, and meanwhile, the precise manipulation of external stimulus would lead to a sophisticated release of cargo drugs from nanocarriers. In addition, the strategy to combine those drug delivery systems which are based on tumor microenvironment or external stimuli with other anticancer drugs or immunoregulatory agents might display synergistic effects in lung cancer therapy.

Overall, the path to find the most suitable stimuli-responsive drug delivery systems for lung cancer therapy still requires numerous contributions in the near future.

## Figures and Tables

**Figure 1 molecules-27-00948-f001:**
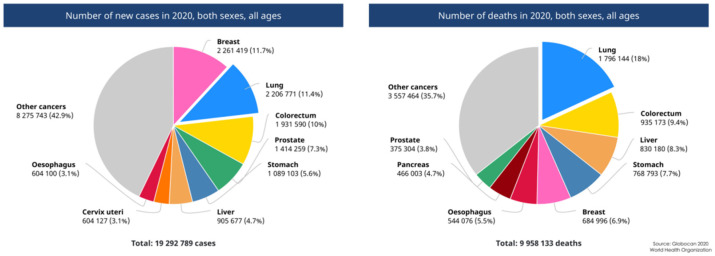
The number of new cases and deaths from cancer worldwide in 2020.

**Figure 2 molecules-27-00948-f002:**
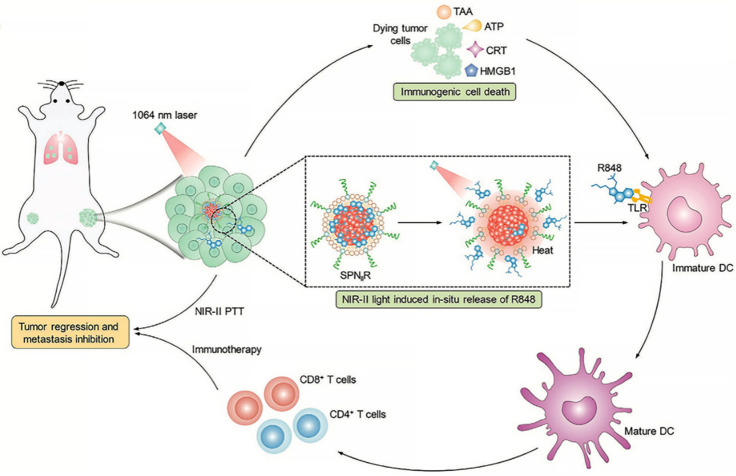
NIR-II light-responsive semiconducting polymer nanoadjuvant (SPNIIR) is designed and applied for synergetic photothermal immunotherapy, not only to the primary and distant tumors but also the metastasis in the lung [37]. PTT, photothermal therapy; TLR, toll-like receptor; DC, dendritic cell; R848, a TLR agonist; TAA, tumor-associated antigens; ATP, adenosine triphosphate; CRT, calreticulin; HMGB1, high mobility group box 1 protein.

**Figure 3 molecules-27-00948-f003:**
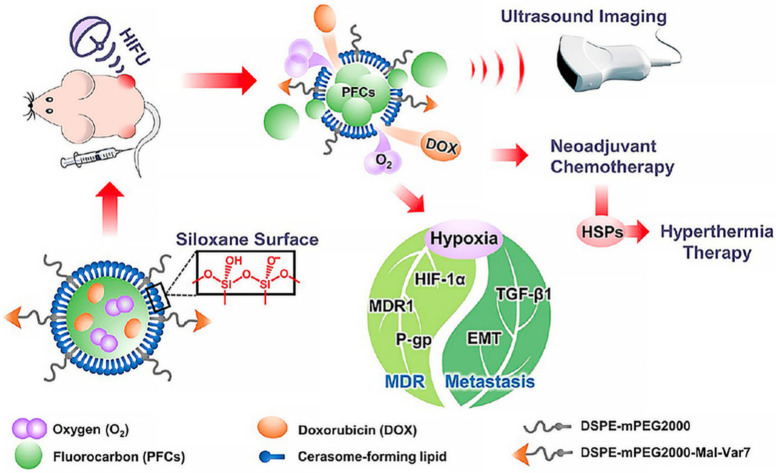
Ultrasound-responsive nanodroplets are designed, fabricated, and capable of inhibiting tumor metastasis in the lung [56]. HIFU, high intensity focused ultrasound; MDR, multi-drug resistance; EMT, epithelial-mesenchymal transition; P-gp, P-glycoprotein; TGF-β1, Transforming Growth Factor-β1; HSPs, heat-shock proteins.

**Figure 4 molecules-27-00948-f004:**
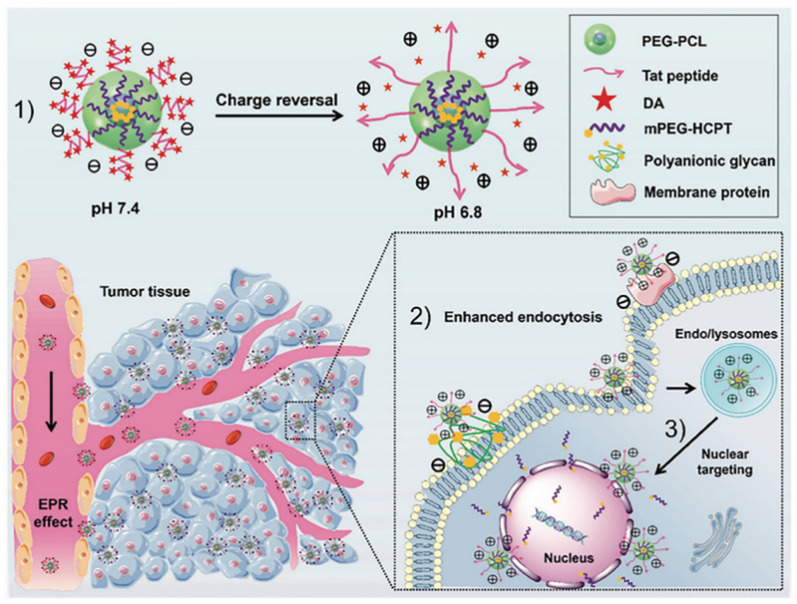
The concept of pH-triggered cell penetration and nuclear targeting for effective cancer therapy for lung metastatic lung cancer. (1) The schematic diagram of charge reversal; (2) The schematic illustration of targeted transport and enhanced uptake of nanoparticles. (3) Nuclear targeting of positively charged nanoparticles [64].

**Table 1 molecules-27-00948-t001:** Brief information of stimuli-responsive nanocarriers that is discussed in this review.

Stimuli	Specific Conditions	Nanocarriers	Diagnostic /Imaging	Therapeutics	Reference
Light	Near-infrared (NIR) light	Gold nanocage@manganese dioxide (AuNC@MnO_2_) nanoparticles	√	√	Lee et al., 2019
		Titania-coated gold nanobipyramids		√	Chen et al., 2019
		Poly(l-lysine)-conjugated chlorin e6 (Ce6) derivative nanoparticle		√	Zhang et al., 2020
		Palladium nanosheet (PdNS)		√	Wang et al., 2018
		Semiconducting polymer nanoadjuvant (SPN_I_IIR)		√	Li et al., 2021
		CE7Q/CQ/S	√	√	Li et al., 2020
	Short-wavelength and NIR light	*O*-nitrobenzyl ester modified polymersome with up-conversion nanoparticles		√	Tsai et al., 2021
Ultrasound	Mechanical effect	Perfluoropentane containing nanobubbles		√	Baspinar et al., 2019
	Chemical effect	PEGylated Co_2_Fe_2_O_4_ nanoflowers (CFP)	√	√	Fu et al., 2021
	Synergistically therapeutic modality	Cerasomal perfluorocarbon nanodroplet (D-vPCs-O_2_)	√	√	Ma et al., 2020
		EXO-DVDMS	√	√	Liu et al., 2019
		Liposome-based nanobubbles	√	√	Lee et al., 2017;
pH	pH 5.7–6.9	CHEMS-based liposomes; HA-ERL/BEV-LPH nanoparticles; DOX-loaded mixed micelles; DA-TAT carrier; mPEG-PHis-PSD; U11-DOX/CUR nanoparticles; *Cis*-aconitic anhydride-modified doxorubicin		√	Park et al., 2021; Pang et al., 2020; Shih et al., 2020; Jing et al., 2018; Shi et al., 2018; Hong et al., 2019; Xia et al., 2018
Enzyme	MMP-2	Cur-P-NPs		√	Han et al., 2017
	MMP-9	MMP-9-sensitive nanocarrier		√	Sidi et al., 2019
	MMPs	A smart inhalable nanocarrier		√	Vaghasiya et al., 2021
	HAase	HPGBCA		√	Ren et al., 2019
	Esterase	Gold nanorod–curcumin conjugate, HAPBA		√	Zhu et al., 2018; Lee et al., 2019
	NQO1	QPA-P		√	Park et al., 2021

√ Nanocarriers that designed for diagnostic/therapeutic applications.

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
