# Peer review of "Stimuli-Responsive Drug Delivery Systems for the Diagnosis and Therapy of Lung Cancer"

_molecules, 2022, doi:10.3390/molecules27030948_

Round 1

Reviewer 1 Report

The review reported by Zhang et al. describes the stimuli responsive smart carriers for for lung cancer therapy. The review is highly recommended for publication after a minor revision.

  1. The authors are requested to add fact sheet data from WHO.
  2. The authors are requested to change et al to et al. throughout the manuscript.
  3. In vivo and in vitro to be changed to in vivo and in vitro (Italic) throughout the manuscript.
  4. The authors are requested to give enough concentration during the preparation of the manuscript (Line 186,187, 190,191,192, 200, 201, 238).
  5. Line 209: Please add space between et and al.
  6. I really don’t understand the meaning of L SEP (Line 345 and 346). Please clarify.
  7. Please remove dot from et al.. (Line 348).
  8. References: Please change the format: The volume will be always in italic.

Author Response

Reviewer(s)' Comments to Author: 

(Reviewer 1) The review reported by Zhang et al. describes the stimuli responsive smart carriers for for lung cancer therapy. The review is highly recommended for publication after a minor revision.

(1) The authors are requested to add fact sheet data from WHO.  

Response: A scheme that described the incidence and death of lung cancer from WHO was included as Figure 1.

  • The authors are requested to change et al to et al. throughout the manuscript.

Response: Thanks for the comment. In the revised manuscript, et al has been changed to et al.

(3) In vivo and in vitro to be changed to in vivo and in vitro (Italic) throughout the manuscript.

Response: Thanks for the comment. In the revised manuscript, in vivo and in vitro have been changed to Italic form.

  • The authors are requested to give enough concentration during the preparation of the manuscript (Line 186,187, 190,191,192, 200, 201, 238).

Response: Thanks for the comment. Some errors occurred during the format transformation of this manuscript, and now these mistakes have been corrected in the revised manuscript.

(5) Line 209: Please add space between et and al.

Response: Thanks for the comment. We have corrected this mistake in the revised manuscript.

  • I really don’t understand the meaning of L SEP (Line 345 and 346). Please clarify.

Response: Thanks for the comment. Some errors occurred during the format transformation of this manuscript, and “L SEP” have been deleted in the revised manuscript.

(7) Please remove dot from et al.. (Line 348).

Response: Thanks for the comment. We have corrected this mistake in the revised manuscript.

(8) References: Please change the format: The volume will be always in italic.

Response: Thanks for the comment. We have corrected the format of references according to the requirement of this journal.

Reviewer 2 Report

This manuscript covers and interesting topic that is suitable for molecules. In general, the review is well written. However, some aspects must be addressed before its publication.

Minor errors:

  • Line 23: Lung instead of lung
  • Line 37: In previous studies? I suggest eliminating this part as there are mechanisms of nanomedicine accumulation
  • Line 59: light-responsive nanocarriers instead of light-responsivenanocarriers
  • Figure 1: please, define abbreviations
  • Discussion: what seems to be the most promising stimuli responsive systems? The authors can discuss this.

Mayor aspects:

  • Did the authors plan to divide each section in diagnosis and therapeutics? I think this could help. Maybe, including tables on this regard may facilitate.
  • The authors should include the stimuli responsive systems that have reached clinic. e.g Thermodox (thermosensitive liposomes containing DOX under clinical trials for hepatic and breast cancer), LiplaCis (liposomes responsive to phospholipase A2) and I think that some iron nanoparticles. They must be mentioned and explained in introduction. 

Author Response

(Reviewer 2)This manuscript covers and interesting topic that is suitable for molecules. In general, the review is well written. However, some aspects must be addressed before its publication.

Minor errors:

  • Line 23: Lung instead of lung

Response: Thanks for the comment. We have corrected this mistake in the revised manuscript.

  • Line 37: In previous studies? I suggest eliminating this part as there are mechanisms of nanomedicine accumulation

Response: Thanks for the comment. This part has been eliminated in the revised manuscript.

  • Line 59: light-responsive nanocarriers instead of light-responsivenanocarriers

Response: Thanks for the comment. We have corrected this mistake in the revised manuscript.

  • Figure 1: please, define abbreviations

Response: Thanks for the comment. We have explained the abbreviations in the legend of Figure 1.

(5) Discussion: what seems to be the most promising stimuli responsive systems? The authors can discuss this.

Response: Thanks for the comment. We have discussed the promising strategies in stimuli-responsive nanocarriers in the discussion section.

Mayor aspects: 

(6) Did the authors plan to divide each section in diagnosis and therapeutics? I think this could help. Maybe, including tables on this regard may facilitate. 

Response: Thanks for the comment. A new table, which briefly included these nanocarriers in this review, has been added in the revised manuscript.

(7) The authors should include the stimuli responsive systems that have reached clinic. e.g Thermodox (thermosensitive liposomes containing DOX under clinical trials for hepatic and breast cancer), LiplaCis (liposomes responsive to phospholipase A2) and I think that some iron nanoparticles. They must be mentioned and explained in introduction.

Response: Thanks for the comment. It is important to discuss the stimuli responsive systems that currently under clinical trials, and these nanocarriers would enlighten us with proper strategies for rational design in the process of nanosystem development. Therefore, we have added a paragraph into the discussion section in the revised manuscript.

Round 2

Reviewer 2 Report

The authors ahve adressed all my presvious comments and improved the manuscript. It deserves to be published in the current, revised, form. 

Author Response

Response to the comment

All reviewer comments/concerns seem to have been addressed in the current version, except R1Q1: "(1) The authors are requested to add fact sheet data from WHO. / Response: A scheme that described the incidence and death of lung cancer from WHO was included as Figure 1."

I am not able to locate any Figure 1, while the original figures have been renumbered in the figure legends (so the previous Fig 1 is now Fig 2 etc.) but NOT in the text, so the text referralas are also now wrongly numbered. There is also no text referral to the allegedly added new Fig 1, which there of course should be if a new figure was added.

When this is fixed, the manuscript can be accepted for publication.

Response: Thank you for pointing out this mistake. In the revised manuscript, all figures have been renumbered and cited in a correct order.
